# KNN Transformer with Pyramid Prompts for Few-Shot Learning

## ABSTRACT

Few-Shot Learning (FSL) aims to recognize new classes with limited labeled data. Recent studies have attempted to address the challenge of rare samples with textual prompts to modulate visual features. However, they usually struggle to capture complex semantic relationships between textual and visual features. Moreover, vanilla self-attention is heavily affected by useless information in images, severely constraining the potential of semantic priors in FSL due to the confusion of numerous irrelevant tokens during interaction. To address these aforementioned issues, a $k$-NN Transformer with Pyramid Prompts (KTPP) is proposed to select discriminative information with $k$-NN Context Attention (KCA) and adaptively modulate visual features with Pyramid Cross-modal Prompts (PCP). First, for each token, the KCA only selects the $k$ most relevant tokens to compute the self-attention matrix and incorporates the mean of all tokens as the context prompt to provide the global context in three cascaded stages. As a result, irrelevant tokens can be progressively suppressed. Secondly, pyramid prompts are introduced in the PCP to emphasize visual features via interactions between text-based class-aware prompts and multi-scale visual features. This allows the ViT to dynamically adjust the importance weights of visual features based on rich semantic information at different scales, making models robust to spatial variations. Finally, augmented visual features and class-aware prompts are interacted via the KCA to extract class-specific features. Consequently, our model further enhances noise-free visual representations via deep cross-modal interactions, extracting generalized visual representation in scenarios with few labeled samples. Extensive experiments on four benchmark datasets demonstrate significant gains over the state-of-the-art methods, especially for the 1-shot task with 2.28% improvement on average due to semantically enhanced visual representations.

## CCS CONCEPTS

• **Computing methodologies → Object recognition**.

## KEYWORDS

Multi-Modality, Prompt, Transformer, Few-Shot Learning

## 1 INTRODUCTION

Deep learning has achieved significant success in computer vision. However, it typically relies heavily on large-scale labeled datasets, which can be costly and sometimes impractical to obtain, such as in rare disease diagnosis [33, 41] and industrial anomaly detection

*ACM MM, 2024, Melbourne, Australia*

© 2024 Copyright held by the owner/author(s). Publication rights licensed to ACM.
ACM ISBN 978-x-xxxx-xxxx-x/YY/MM
https://doi.org/10.1145/nnnnnnn.nnnnnnn

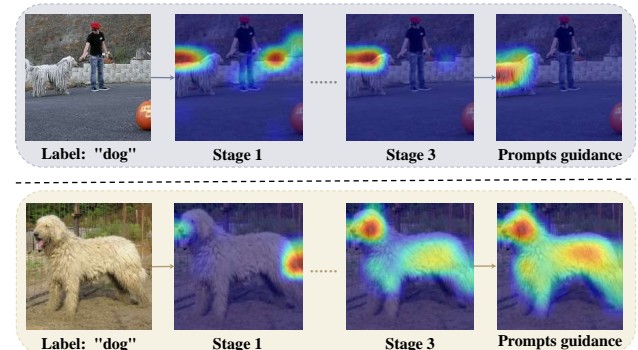

**Figure 1: The image annotated as "dog" contains abundant spurious information such as people, walls, etc. Moreover, the scale variations present across different images, thus limiting the performance of ViTs in FSL. Our KTPP achieves coarse-to-fine filtration of noise, adaptation to spatial variations, and prompts-guided class-specific visual extraction.**

[15, 25]. Instead, humans can recognize new objects from only a few images. To emulate human learning ability and reduce the dependency on annotated data, Few-Shot Learning (FSL) [50] is proposed to learn from a limited number of data, which has drawn considerable concern in recent years. Despite of remarkable progress [1, 7, 13, 14, 20, 50, 60, 68], achieving human-level performance remains challenging, especially in the 1-shot scenario with only one labeled sample.

Most methods [7, 13, 14, 50] utilize support features to infer class prototypes for classification. However, the limited number of labeled samples makes it challenging to learn discriminative features, leading to semantic biases in the generation of class-specific prototypes. To address this issue, an alternative is to involve integrating additional semantic information (e.g., category descriptions) with visual prototypes. Following this view, some studies [6, 31, 61–63, 73] reveal that integration of semantic priors can enhance prototype representation. Despite their success, most methods integrate semantic embeddings at a higher hierarchical level (e.g., prototypes and classifiers), lacking underlying feature interactions and ignoring distribution differences between modalities. Moreover, a recent study [6] attempted to modulate visual features by semantic embeddings of labels as prompts during the extraction process. However, it struggles to capture complex semantic relationships between textual and visual features by employing linear layers for direct and single-scale fusion.

Moreover, several approaches [6, 13, 20] perform interactions within or across modalities in Vision Transformers (ViTs) via vanilla self-attention. However, for vanilla self-attention where each patch is forced to interact with all tokens for computing the attention matrix, noisy tokens are inevitably introduced into the calculation.

This incorporation of noisy tokens renders image features vulnerable to spurious correlations, especially in cluttered backgrounds and occlusions. As shown in Fig. 1, the image annotated as "dog" consists of one person, apple, wall, tree, and other class-irrelevant objects, making the ViT falsely associate these objects with the label when only few labeled samples are available. Therefore, the potential of semantic priors is severely constrained in cross-modal interactions via vanilla self-attention due to the confusion of numerous irrelevant tokens during interaction. As a result, designing an effective cross-modal fusion strategy and improving the self-attention mechanism are crucial to achieving deep cross-modal interactions and filtering out class-irrelevant visual features in FSL frameworks.

To address these aforementioned issues, a $k$-NN Transformer with Pyramid Prompts (KTPP) is proposed to fully leverage semantic information and capture class-specific support features. Specifically, the KTPP mainly consists of two modules: $K$-NN Context Attention (KCA) and Pyramid Cross-modal Prompts (PCP). Firstly, the KCA selects the $k$ most relevant tokens for each token to compute attention weights instead of all tokens, effectively excluding irrelevant information and reducing the computational burden. As a result, irrelevant tokens can be progressively suppressed in a coarse-fine manner via three cascaded stages, as shown in Fig. 1. To avoid the issue of paying too much attention to the local context and neglecting the discriminative information of global feature [19, 32] in the top-$k$ operation, the mean of all tokens is regarded as context prompts which are incorporated to enhance the global context information of each token. Secondly, the pre-trained CLIP model [46] has an excellent capability of cross-modal feature alignments, extracting rich and unbiased semantic embeddings. Therefore, we exploit the class-aware prompts encoded from class names via the CLIP. the PCP enables these class-aware prompts to interact with multi-scale support features, learning pyramid prompts that capture complex semantic relationships between textual and visual features. The pyramid structure allows the ViT to dynamically focus on the support features most relevant to the semantic information, making it robust to spatial variations. Finally, the augmented support features and the class-aware prompts are then interacted via the KCA, which enables the ViT to further capture discriminative support features due to the semantically guided class-specific features. As shown in Fig. 1, the KTPP progressively filters out noisy tokens at different stages and further enhances noise-free visual representations via deep cross-modal interactions guided by prompts.

Overall, our contributions are summarized as follows:

- A KTPP method is proposed to fully exploit cross-modal information and select discriminative visual features.
- A $K$-NN Context Attention module is proposed to filter out class-irrelevant features in a coarse-fine manner, capturing class-specific features in cross-modal interactions and improving computational efficiency.
- Pyramid Cross-modal Prompts are proposed to enhance visual features via deep cross-modal interactions, improving the adaptivity ability to spatial variations.
- The proposed method achieves significant performance in four FSL benchmarks, particularly improving 1-shot accuracy by 1%-4% compared to the state-of-the-art approaches.

## 2 RELATED WORK

### 2.1 Few-Shot Learning

In general, methods for Few-Shot Learning can be divided into two types. One is optimization-based methods [16, 42, 49, 74], aimed at quickly adapting models to new data. The other type is metric-based methods adopted in this work aims to classify by measuring distances between support and query samples within the feature space. ProtoNet [50] uses Euclidean distance. Deepemd [68] utilizes the Earth Mover's Distance. others [12, 21, 36, 64, 71] have integrated Transformer layers as classifiers. Moreover, recent works have incorporated textual modality as auxiliary information to boost generalization on novel classes. AM3 [61] merges semantic and visual prototypes through an adaptive fusion mechanism. Multiple-Semantics [66] aims to enhance prototype representation by integrating multiple semantic information TRAML [31] leverages semantic similarities between classes to improve classification accuracy by a task-relevant adaptive margin loss. SVAE [62] generates additional visual features from semantic information through a Conditional Variational Autoencoder (CVAE) [51], resulting in more reliable prototypes. KTN [45] utilizes graph convolutional networks and knowledge graphs to explicitly leverage semantic information for learning classifiers. These methods apply semantic information at a higher level such as prototypes and classifiers, which lack foundational visual feature interactions and ignore distribution differences. In contrast, we utilize ViT as the backbone, leveraging semantic information in the extraction process to modulate visual features adaptively.

### 2.2 Prompt Learning

To diminish the dependency on extensive training datasets and enhance the flexibility of pre-trained models for downstream tasks, several prompt-based approaches [6, 10, 23, 46, 53, 72, 73] have been proposed. CLIP [46] utilizes constructed prompts like "a photo of a class" to compute similarity scores between image embeddings and textual prompts for classification. CoOp [73] introduces learnable continuous prompt tokens for fine-tuning the pre-trained CLIP. Further building on CoOp, CoCoOp [72] employs a Meta-Net to transform each prompt token into instance-specific contexts, significantly improving model adaptability. MAPLE [23] designs corresponding prompts for each modality to guide the learning of the respective modality. VPT [22] introduces learnable parameters as visual prompts to fine-tune the CLS token of the ViT, enhancing the flexibility and adaptability of the model. Frozen [53] conducts multimodal few-shot learning by using image features paired with text as prompts. SP [6] utilizes textual labeled embedding matching images as prompts to supplement visual features by employing linear layers for direct and single-scale fusion. In this paper, we propose Pyramid Cross-modal Prompts to learn pyramid prompts, which can dynamically adjust the importance weights of visual features based on rich semantic information at different scales, making models robust to spatial variations.

### 2.3 Vision Transformer

Vision Transformers (ViTs) have shown remarkable success in numerous computer vision tasks [2, 5, 26, 34, 37, 43, 59, 70], benefiting

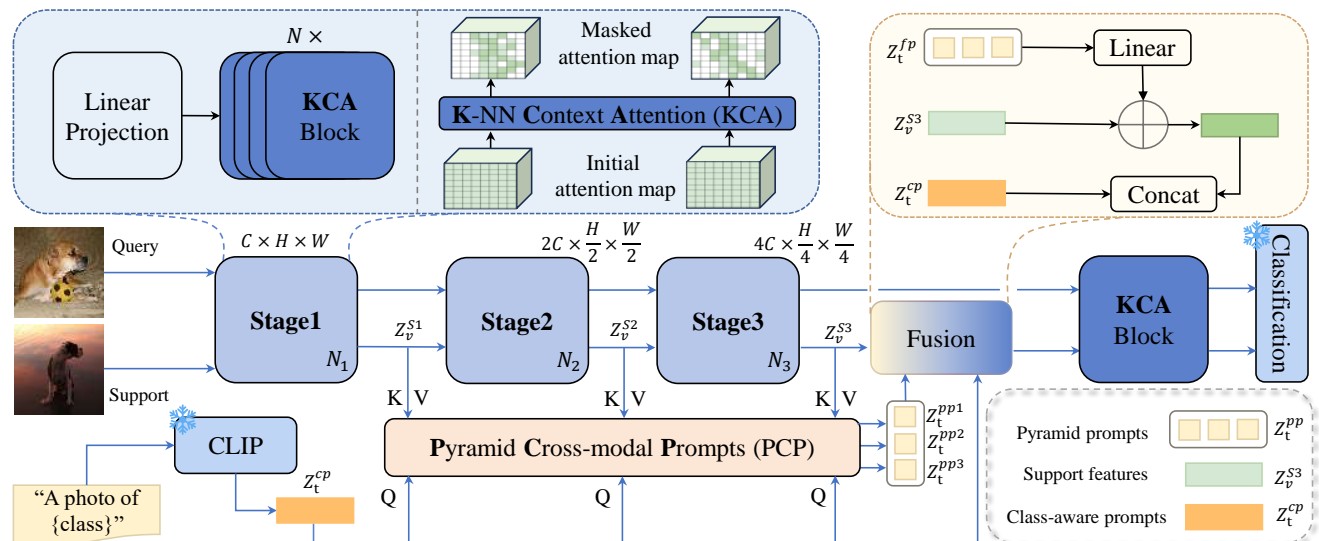

**Figure 2: The framework of our proposed KTPP. Image patches are sequentially fed into three cascaded stages where KCA filters out irrelevant tokens and linear projection extracts multi-scale support features $Z_v^S$. Text-based class-aware prompts $Z_t^{cp}$ are exploited via the CLIP. Pyramid prompts $Z_t^{pp}$ are obtained to enhance support features via the PCA between $Z_t^{cp}$ and $Z_v^S$. The enhanced support features and $Z_t^{cp}$ are interacted via the KCA to extract class-specific support features.**

from their proficiency to capture long-range dependencies among image patches. To match or surpass the performance of CNNs of similar size trained on ImageNet [11], ViTs are usually trained on large-scale datasets. Few recent works [4, 13, 20, 58] have explored the adaptation of ViTs on small datasets. IDMM [4] pretrains ViT using instance discrimination loss, followed by fine-tuning on the target dataset. SUN [13] introduces local supervision to enhance dependency learning of tokens. FewTURE [20] also achieves excellent performance with a fully Transformer-based architecture. SP [6] utilizes semantic prompts to enhance the visual feature by cross-modal interactions based on self-attention. Despite these advances, the fully connected self-attention mechanism in these models usually introduces noisy tokens, which can degrade the quality of the representation. KVT [58] attempts to refine this mechanism, yet the performance is limited in scenarios with few labeled samples due to excessive focus on local feature context. In this paper, we propose a $k$-NN Context Attention mechanism that progressively filters out irrelevant information in visual modality via three cascaded stages, captures discriminative features in cross-modal interactions guided by semantic information, and enhances global context information in FSL. To the best of our knowledge, we are the first to consider the way of integrating cross-modal data and context prompts for improving $k$-NN attention and applying it in FSL.

## 3 METHOD

We first overview the whole pipeline, and then introduce the preliminary of FSL. Next, two novel components of K-NN Context Attention (KCA) and Pyramid Cross-modal Prompts (PCP) are presented.

### 3.1 Overview

Fig. 2 shows the framework of our proposed KTPP. To suppress irrelevant information, the KCA progressively filters out noisy tokens in three cascaded stages. To exploit rich semantic priors, class-aware prompts are extracted by inputting class names into the CLIP. To capture complex semantic relationships between different modalities, pyramid prompts are obtained to enhance support features by applying the PCP at different scales, enabling the ViT to dynamically adjust the importance weights of support features based on semantic information. Consequently, models are robust to spatial variations. To select discriminative support features, enhanced support features and class-aware prompts are interacted via the KCA, which can extract class-specific support features guided by semantic information. Finally, the ViT can generate generalized class prototypes for classification.

### 3.2 Preliminary

Few-Shot Learning (FSL) focuses on generalizing the knowledge learned from training classes $C_{\text{train}}$ to test classes $C_{\text{test}}$ with only a few labeled samples, where the training and test categories are disjoint ($C_{\text{train}} \cap C_{\text{test}} = \emptyset$). This task is typically formalized as an N-way M-shot problem where the training involves N distinct categories and each category has M labeled images. An episodic training strategy [55] is adopted, where each episode consists of two parts: a support set $S = \{(x_i, y_i)\}_{i=1}^{N \times M}$ with $N \times M$ labeled samples, and a query set $Q = \{(x_i, y_i)\}_{i=1}^{N \times H}$ with $N \times H$ test samples to evaluate the performance.

In the FSL approach, prototype-based inference is commonly utilized. Specifically, N class prototypes are obtained by averaging

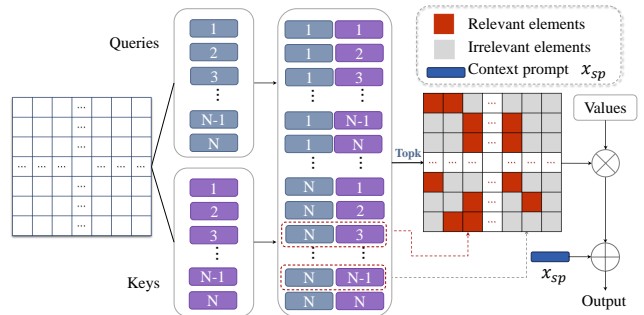

**Figure 3: Illustration of K-NN Context Attention (KCA). For each query, the $k$ highest scorers from the N query-key pairs are selected for computing attention weights, and the rest are set to negative infinity. The context prompt is the mean of all tokens weighted by the KCA.**

all samples of each class as follows:

$$c_i = \frac{1}{|S_i|} \sum_{j=1}^{M} f(x_j) , \quad (1)$$

where $x_j \in S_i$. $S_i$ denotes the i-th support class. $|S_i|$ represents the number of samples in each support class. $f$ is the feature extractor.

For a query image $q_i$, the similarity score between $q_i$ and all support classes is determined by calculating the distance function between $q_i$ and N class prototypes. The probability of $q_i$ belonging to the n-th support class is calculated as follows:

$$p_k = \frac{\exp(\gamma \cdot d(f(q_i), c_i))}{\sum_{j=1}^{N} \exp(\gamma \cdot d(f(q_i), c_j))} , \quad (2)$$

where $\gamma$ is a temperature parameter. $d()$ denotes the distance function. The support class corresponding to the maximum probability is regarded as the classification result.

## 3.3  $k$-NN Context Attention

Following[55], the vanilla self-attention formula is defined as:

$$\hat{V} = \text{softmax}\left(\frac{QK^T}{\sqrt{d}}\right)V. \quad (3)$$

The fully connected self-attention mechanism allocates weights to all tokens without distinguishing noise, making ViTs susceptible to noisy tokens when only a limited number of samples are available. To mitigate this issue, we propose a $K$-NN Context Attention (KCA) mechanism, which aims to progressively filter out irrelevant tokens in a coarse-fine manner via three cascaded stages.

Specifically, the KCA computes the attention matrix $A \in \mathbb{R}^{n \times n}$ by the dot product of all query-key pairs, like vanilla attention:

$$A = \frac{QK^T}{\sqrt{d}}. \quad (4)$$

We further construct a masked attention matrix $\hat{A} \in \mathbb{R}^{n \times n}$, selecting the $k$ most relevant keys for each query, as shown in Fig. 3. This is achieved by performing the top-$k$ operation on each row of the attention matrix to obtain the highest $k$ scores and their indexes.

In the $\hat{A}$, the selected elements remain unchanged and all others are set to negative infinity. The formulation is defined as follows:

$$\hat{A}_{ij} = \text{Mask}_{k-nn}(A_{ij}) = \begin{cases} A_{ij} & \text{if } (i, j) \text{ in } I \in \mathbb{R}^{n \times k} \\ -\infty & \text{otherwise} \end{cases}, \quad (5)$$

where $I \in \mathbb{R}^{n \times k}$ represents the collection of $n \times k$ indexes.

Therefore, the KCA naturally forms a sparse attention matrix, concentrating the focus of the ViT on class-relevant features. In the proposed framework, image patches are sequentially fed into three stages to perform the KCA, thereby progressively filtering out irrelevant tokens by selecting similar tokens and refining features in a coarse-fine manner.

In vanilla self-attention, attention weights are calculated based on the relationships between the position and all other positions, resulting in $O(N^2)$ computational complexity for a sequence of length $N$. In contrast, the KCA considers only the $K$ nearest neighbors for each position, reducing computational complexity to approximately $O(NK)$, where $K$ is much smaller than $N$. This approximation significantly reduces computational costs and maintains ins relatively high performance.

However, a potential risk is the over-focus on local features to the detriment of global context. This limits the generalization of the ViT to new classes with few samples. To address this problem, an average representation of the entire tokens is used as the context prompt to capture global information:

$$z_{cp} = \text{Average}(\text{softmax}(\hat{A}_{ij})V), \quad (6)$$

where $z_{cp}$ represents context prompts. Incorporating $z_{cp}$ into each token can enhance global context information.

The whole KCA formula can be defined as:

$$\tilde{V} = \text{softmax}(\text{Mask}_{k-NN}(\frac{QK^T}{\sqrt{d}}))V + \alpha z_{cp}, \quad (7)$$

where $\alpha$ is a learnable parameter, allowing the ViT to adaptively adjust the coefficient between local and global context information.

## 3.4  Pyramid Cross-modal Prompts

The proposed Pyramid Cross-modal Prompts (PCP) leverage semantic embeddings to enhance support representations, allowing the ViT to adaptively modulate support features based on semantic information, capturing complex semantic correlations between textual and visual modalities via deep cross-modal interactions. The key component of PCP is the cross-modal enhancement module which is illustrated in Fig. 4. It illustrates cross-modal attention to leverage semantic information and generate semantic-based visual features.

Specifically, to exploit rich and unbiased semantic information, class-aware prompts $Z_t^{cp} \in \mathbb{R}^{L_t \times d_t}$ from class names are obtained by the CLIP which has an excellent capability of cross-modal feature alignments. Additionally, three support features $Z_v^{Si} \in \mathbb{R}^{L_v \times d_v}$ at different scales are extracted via three stages.

Given weight matrices $W_{Q_t} \in \mathbb{R}^{d_t \times d}$, $W_{K_v} \in \mathbb{R}^{d_v \times d}$, and $W_{V_v} \in \mathbb{R}^{d_v \times d}$, the query $Q$, key $K$, and value $V$ in the cross-modal attention

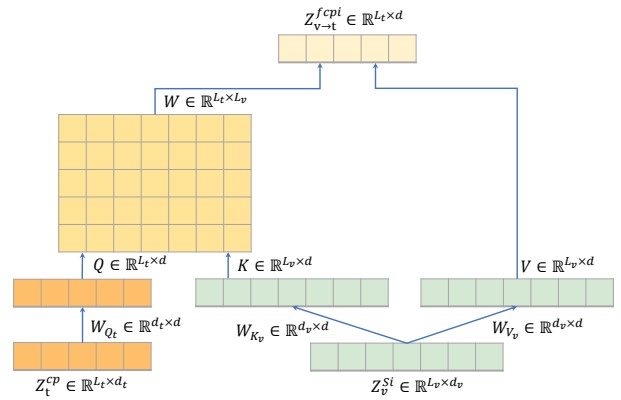

**Figure 4: Illustration of cross-modal enhancement module in the PCA. Class-aware prompts $Z_t^{cp}$ provide queries $Q$, dynamically adjusting the importance weights of visual features.**

are computed as follows:

$$Q = Z_t^{cp} W_{Q_t} \in \mathbb{R}^{L_t \times d},$$
$$K = Z_v^{Si} W_{K_v} \in \mathbb{R}^{L_v \times d}, \qquad (8)$$
$$V = Z_v^{Si} W_{V_v} \in \mathbb{R}^{L_v \times d}.$$

Next, the attention weight matrix of textual modality for visual modality is obtained as:

$$W_{v \to t} = \text{softmax}(QK^T \beta), \qquad (9)$$

where $W_{v \to t}$ is the attention weight matrix and $\beta$ is a scaling parameter. Due to discrepancies in modality distributions, the initial training phase may yield limited relevance between textual and visual features, resulting in small values within the weight matrix. To enhance the learning of model parameters, we introduce the hyperparameter $\beta$ to scale the matrix before applying softmax.

After applying the identical operation to the support features of the remaining two scales, we derive the pyramid prompts $Z_{v \to t}^{fpp}$, representing the inherent semantic details of visual features chosen by textual features across various levels. Utilizing the multi-scale operation, ViT captures diverse spatial information, thereby enhancing its capability to adapt to spatial variations.

The pyramid prompts are linearly mapped to the dimension of the support feature $Z_v^{S3}$ for fusion, defined as:

$$\widetilde{Z_v^S} = Z_v^{S3} + \lambda \sum_{i=1}^{N} \text{Linear}(Z_{v \to t}^{ppi}), \qquad (10)$$

where $\widetilde{Z_v^S}$ represents the support features emphasized by the pyramid prompts, $N$ is typically set to 3, and $\lambda$ as a learnable parameter.

The augmented support features $\widetilde{Z_v^S}$ and class-aware prompts $Z_t^{cp}$ are interacted in the ViT via the KCA. The output $y_v^S$ is calculated as follows:

$$y_v^S = KCA\left[\left(\widetilde{Z_v^S} \| Z_t^{cp}\right)\right], \qquad (11)$$

where $\|$ represents the concatenation operation along the spatial dimension. The KCA enables the ViT to extract discriminative

support features due to the semantically guided capture of class-specific features, followed by obtaining more generalized class prototypes for classification via Eq. 1.

The similarity scores between the query samples and class prototypes are computed via the cosine distance, defined as follows:

$$p_k = \frac{\exp(\gamma \cdot \cos(f(q_i), c_i))}{\sum_{j=1}^{N} \exp(\gamma \cdot \cos(f(q_i), c_j))}, \qquad (12)$$

where $\cos()$ denotes the cosine distance function. The classification results of the query image $q_i$ is the support class with maximum probability.

The ViT is trained by minimizing the Cross-Entropy (CE) loss function, which can be formulated as follows:

$$\Gamma = \arg \min \sum_{i=1}^{M} \mathcal{L}^{CE}(f(q_i), y_i), \qquad (13)$$

where $\mathcal{L}^{CE}$ represents CE loss. $f()$ represents the ViT backbone. $q_i$ and $y_i$ donate a query image and its corresponding ground-truth label.

## 4 EXPERIMENTS

We first present the four benchmark datasets. Next, we introduce the details of the experiments, followed by the few-shot classification performance and the model analysis.

### 4.1 Datasets

The effectiveness of the proposed method is evaluated on four few-shot datasets: miniImageNet [56], tieredImageNet [47], CIFAR-FS [30], and FC100 [44]. MiniImageNet and TieredImageNet are both subsets of ImageNet [11]. MiniImageNet has 64 training classes, 16 validation classes, and 20 test classes. TieredImageNet includes 351 training classes, 97 validation classes, and 160 test classes. CIFAR-FS and FC100 are derived from the CIFAR-100 dataset [27]. CIFAR-FS has 64 training classes, 16 validation classes, and 20 test classes employing a random partitioning strategy. FC100 adopts a unique superclass partitioning method, featuring 12 superclasses in the training set (equivalent to 60 classes), and 4 superclasses each in the validation and test sets, totaling 20 classes. Notably, only the training set is utilized for model training, with no overlap between the training, validation, and test sets in few-shot learning settings.

### 4.2 Implementation Details

**Backbone**. In all experimental setups, we employ Visformer-T [8] as the feature extractor. The Visformer-T model boasts a more small number of parameters, rendering our model smaller compared to Resnet-12 models [48, 50, 68], or even several times smaller than the ViT-S/16 and WRN-28 models [20, 49, 57]. KCA Block consists of two Batchnorm layers, a two-layer MLP, and Multi-Head KCA. Furthermore, we employ ViT-B/32 CLIP as a text encoder, extracting an output dimension of 512.

**Training Details**. Our approach follows a two-phase training procedure, similar to traditional frameworks such as Meta-baseline [7], including pre-training and meta-tuning stages. Throughout both phases, KCA serves as the attention mechanism for the ViT. Moreover, PCP is employed in the meta-tuning phase. For input

**Table 1: Results (%) for the 5-way 1-shot and 5-way 5-shot settings on MiniImageNet and TieredImageNet. The average accuracy with 95% confidence interval are reported. Bold font indicates the best results. Blue font indicates the suboptimal results.**

| Methods | Venue | Backbone | Params | MiniImageNet 5-way | | TieredImageNet 5-way | |
|---|---|---|---|---|---|---|---|
| | | | | 1-shot | 5-shot | 1-shot | 5-shot |
| ProtoNet [50] | NeurIPS'2017 | ResNet-12 | 12.4M | 62.39 ± 0.21 | 80.53 ± 0.14 | 68.23 ± 0.23 | 84.03 ± 0.16 |
| KTN [45] | ICCV'2019 | ResNet-12 | 12.4M | 61.42 ± 0.72 | 74.16 ± 0.56 | - | - |
| AM3 [61] | NeurIPS'2019 | ResNet-12 | 12.4M | 65.30 ± 0.49 | 78.10 ± 0.36 | 69.08 ± 0.47 | 82.58 ± 0.31 |
| TRAML [31] | CVPR'2020 | ResNet-12 | 12.4M | 67.10 ± 0.52 | 79.54 ± 0.60 | - | - |
| DeepEMD [68] | CVPR'2020 | ResNet-12 | 12.4M | 65.91 ± 0.82 | 82.41 ± 0.56 | 71.16 ± 0.87 | 86.03 ± 0.58 |
| Meta-Baseline [7] | ICCV'2021 | ResNet-12 | 12.4M | 63.17 ± 0.23 | 79.26 ± 0.17 | 68.62 ± 0.27 | 83.29 ± 0.18 |
| DeepBDC [60] | CVPR'2022 | ResNet-12 | 12.4M | 67.34 ± 0.43 | 84.46 ± 0.28 | 72.34 ± 0.49 | 87.31 ± 0.32 |
| SVAE [62] | CVPR'2022 | ResNet-12 | 12.4M | 74.84 ± 0.23 | 83.28 ± 0.40 | 76.98 ± 0.65 | 85.77 ± 0.50 |
| Meta-AdaM [52] | NeurIPS'2023 | ResNet-12 | 12.4M | 59.89 ± 0.49 | 77.92 ± 0.43 | 65.31 ± 0.48 | 85.24 ± 0.35 |
| ProtoDiff [14] | NeurIPS'2023 | ResNet-12 | 12.4M | 66.63 ± 0.21 | 83.48 ± 0.15 | 72.95 ± 0.24 | 85.15 ± 0.18 |
| ESPT [48] | AAAI'2023 | ResNet-12 | 12.4M | 68.36 ± 0.19 | 84.11 ± 0.12 | 72.68 ± 0.22 | 87.49 ± 0.14 |
| FGFL [9] | ICCV'2023 | ResNet-12 | 12.4M | 69.14 ± 0.80 | 86.01 ± 0.62 | 73.21 ± 0.88 | 87.21 ± 0.61 |
| BMI [35] | MM'2023 | ResNet-12 | 12.4M | 72.96 ± 0.36 | 81.94 ± 0.29 | 75.45 ± 0.44 | 84.77 ± 0.33 |
| 4S [40] | MM'2023 | ResNet-12 | 12.4M | 74.53 ± 0.68 | 85.78 ± 0.49 | - | - |
| MetaDiff [67] | AAAI'2024 | ResNet-12 | 12.4M | 64.99 ± 0.77 | 81.21 ± 0.56 | 72.33 ± 0.92 | 86.31 ± 0.62 |
| ALFA [3] | TPAMI'2024 | ResNet-12 | 12.4M | 66.61 ± 0.28 | 81.43 ± 0.25 | 70.29 ± 0.40 | 86.17 ± 0.35 |
| LastShot [65] | TPAMI'2024 | ResNet-12 | 12.4M | 67.35 ± 0.20 | 82.58 ± 0.14 | 72.43 ± 0.23 | 85.82 ± 0.16 |
| LEO [49] | ICLR'2019 | WRN-28 | 36.5M | 61.76 ± 0.08 | 77.59 ± 0.12 | 66.33 ± 0.05 | 81.44 ± 0.09 |
| Align [1] | ECCV'2022 | WRN-28 | 36.5M | 65.92 ± 0.60 | 82.85 ± 0.55 | 74.40 ± 0.68 | 86.61 ± 0.59 |
| AMTNet [28] | MM'2022 | WRN-28 | 36.5M | 70.05 ± 0.46 | 84.55 ± 0.29 | 73.86 ± 0.50 | 87.62 ± 0.33 |
| RankDNN [18] | AAAI'2023 | WRN-28 | 36.5M | 66.67 ± 0.15 | 84.79 ± 0.11 | 74.00 ± 0.15 | 88.80 ± 0.25 |
| STANet [29] | AAAI'2023 | WRN-28 | 36.5M | 69.86 ± 0.46 | 85.16 ± 0.29 | 74.41 ± 0.50 | 87.64 ± 0.33 |
| SUN [13] | ECCV'2022 | Visformer-S | 12.4M | 67.80 ± 0.45 | 83.25 ± 0.30 | 72.99 ± 0.50 | 86.74 ± 0.33 |
| FewTURE [20] | NeurIPS'2022 | ViT-S/16 | 22.0M | 68.02 ± 0.88 | 84.51 ± 0.53 | 72.96 ± 0.92 | 86.43 ± 0.67 |
| SP [6] | CVPR'2023 | Visformer-T | 10.3M | 72.31 ± 0.40 | 83.42 ± 0.30 | 78.03 ± 0.46 | 88.55 ± 0.32 |
| KTPP | ours | Visformer-T | 11.1M | **76.71 ± 0.37** | **86.46 ± 0.27** | **80.80 ± 0.43** | **90.01 ± 0.29** |

images with a resolution of 224 × 224, Visformer-T sequentially extracts feature dimensions of 96, 192, and 384. We utilize the AdamW optimizer [39] with a learning rate of 5e-4. During the pre-training phase, the number of training epochs is set to 500 for the miniImageNet, CIFAR-FS, and FC100 datasets, and 300 epochs for the tieredImageNet dataset. A batch size of 512 is employed, and a cosine learning rate scheduler [38] is utilized for learning rate adjustment. In the meta-tuning phase, the model is trained for 100 epochs using the episodic training strategy. All experiments are conducted on an NVIDIA GeForce RTX 3090 GPU, utilizing PyTorch for code implementations.

**Evaluation Protocol**. The proposed method is evaluated in 5-way 1-shot and 5-way 5-shot settings. Each class has 15 test instances. We randomly sampled 2,000 episodes from the test class. The average accuracy with 95% confidence interval is reported.

## 4.3 Comparison with the SOTA Methods

To evaluate the effectiveness of our proposed KTPP, extensive experiments are conducted on four benchmark datasets. Tabs. 1 and 2 present the results of KTPP compared to state-of-the-art methods under 5-way 1-shot and 5-way 5-shot settings.

KTPP demonstrates substantial superiority over the counterparts across all settings. Specifically, in the 1-shot settings, KTPP exhibits

notable advancements, outperforming state-of-the-art results by 1.45% to 3.06% across the four benchmarks. KTPP significantly outperforms previous textual modality-based methods (AM3 [61], KTN [45], TRAML [31], SVAE [62], and SP [6]), improving 1-shot accuracy by 4.4% and 5-shot accuracy by 3.04% on miniImageNet compared to SP. In addition, Compared to previous ViT-based methods (SUN [13], FewTure [20]), KTPP achieves superior results with fewer parameters, improving 1-shot accuracy by 7.81% and 5-shot accuracy by 3.27% on TieredImageNet.

The effectiveness of KTPP is confirmed by the results. This is attributed to the ability of KTPP to progressively filter out class-irrelevant features in a coarse-fine manner and capture discriminative in cross-modal interactions via the KCA. Moreover, pyramid prompts learned in the PCP contribute to enhancing visual features through deep cross-modal interactions, improving adaptability to spatial variations via the pyramid structure.

## 4.4 Model Analysis

### 4.4.1 *Ablation Study*. 
To fairly prove the effectiveness of our method, we leverage the ViT as the baseline. KCA and PCP are introduced to investigate their impact, as shown in Table 3. First KCA is integrated as the attention mechanism into the baseline, improving the 1-shot and 5-shot accuracy by 3.25% and 2.89% in

**Table 2: Results (%) for the 5-way 1-shot and 5-way 5-shot settings on CIFAR-FS and FC100. The average accuracy with 95% confidence interval are reported. Bold font indicates the best results. Blue font indicates the sub-optimal results.**

| Methods | Venue | Backbone | Params | CIFAR-FS 5-way | | FC100 5-shot | |
|---------|-------|----------|--------|------------------|------------------|------------------|------------------|
| | | | | 1-shot | 5-shot | 1-shot | 5-shot |
| ProtoNet [50] | NeurIPS'2017 | ResNet-12 | 12.4M | 72.20 ± 0.70 | 83.50 ± 0.50 | 41.54 ± 0.76 | 57.08 ± 0.76 |
| TADAM [44] | NeurIPS'2018 | ResNet-12 | 12.4M | - | - | 40.10 ± 0.40 | 56.10 ± 0.40 |
| MABAS [24] | ECCV'2020 | ResNet-12 | 12.4M | 72.80 ± 0.70 | 84.30 ± 0.50 | 47.20 ± 0.60 | 55.50 ± 0.60 |
| Meta-NVG [69] | ICCV'2021 | ResNet-12 | 12.4M | 73.51 ± 0.92 | 85.65 ± 0.65 | 42.31 ± 0.75 | 58.16 ± 0.78 |
| Meta-AdaM [52] | NeurIPS'2023 | ResNet-12 | 12.4M | - | - | 41.12 ± 0.49 | 56.14 ± 0.49 |
| ALFA [3] | TPAMI'2024 | ResNet-12 | 12.4M | 76.32 ± 0.43 | 86.73 ± 0.31 | 44.54 ± 0.50 | 58.44 ± 0.42 |
| LastShot [65] | TPAMI'2024 | ResNet-12 | 12.4M | 76.76 ± 0.21 | 87.49 ± 0.12 | 44.08 ± 0.18 | 59.14 ± 0.18 |
| PN+rot [17] | ICCV'2019 | WRN-28 | 36.5M | 69.55 ± 0.34 | 82.34 ± 0.24 | - | - |
| Align [1] | ECCV'2022 | WRN-28 | 36.5M | - | - | 45.83 ± 0.48 | 59.74 ± 0.56 |
| AMTNet [28] | MM'2022 | WRN-28 | 36.5M | 80.38 ± 0.48 | 89.89 ± 0.32 | - | - |
| SUN [13] | ECCV'2022 | Visformer-S | 12.4M | 78.37 ± 0.46 | 88.84 ± 0.32 | - | - |
| FewTURE [20] | NeurIPS'2022 | ViT-S/16 | 22.0M | 72.80 ± 0.88 | 86.14 ± 0.64 | 46.20 ± 0.79 | 63.14 ± 0.73 |
| SP [6] | CVPR'2023 | Visformer-T | 10.3M | 82.18 ± 0.40 | 88.24 ± 0.32 | 48.53 ± 0.38 | 61.55 ± 0.41 |
| KTPP | ours | Visformer-T | 11.1M | **83.63 ± 0.57** | **90.19 ± 0.30** | **51.59 ± 0.40** | **65.18 ± 0.40** |

**Table 3: Ablation study of KTPP on miniImageNet. "KCA" means K-NN Context Attention. "PCP" denotes Pyramid Cross-modal Prompts.**

| Training phase | | Module | | 1-shot | 5-shot |
|------|------|------|------|--------|--------|
| Pre | Meta | KCA | PCP | | |
| ✓ | | | | 63.84 ± 0.38 | 80.72 ± 0.31 |
| ✓ | ✓ | | | 65.56 ± 0.37 | 81.62 ± 0.31 |
| ✓ | | ✓ | | 67.09 ± 0.37 | 83.61 ± 0.30 |
| ✓ | ✓ | ✓ | | 68.75 ± 0.38 | 84.92 ± 0.29 |
| ✓ | ✓ | | ✓ | 73.80 ± 0.38 | 84.34 ± 0.28 |
| ✓ | ✓ | ✓ | ✓ | **76.71 ± 0.37** | **86.46 ± 0.27** |

**Table 4: The effect of value $k$.**

| The value of $k$ | 1-shot | 5-shot |
|------------------|--------|--------|
| $k = 1$ | 71.55 ± 0.35 | 83.65 ± 0.29 |
| $k = 5$ | 75.13 ± 0.37 | 84.49 ± 0.28 |
| $k = 7$ | 75.50 ± 0.35 | 85.69 ± 0.28 |
| $k = 10$ | **76.71 ± 0.37** | **86.46 ± 0.27** |
| $k = 15$ | 76.21 ± 0.38 | 85.92 ± 0.27 |
| $k = 20$ | 75.81 ± 0.38 | 84.81 ± 0.29 |
| $k = 30$ | 75.01 ± 0.37 | 84.39 ± 0.30 |

is optimal on miniImageNet, which enables the ViT to distinguish class-irrelevant tokens more effectively.

**Table 5: The performance of different backbone architectures. * denotes results produced by our model.**

| method | backbone | 1-shot | 5-shot |
|--------|----------|--------|--------|
| ProtoNet [50] | Resnet-12 | 62.39 ± 0.21 | 80.53 ± 0.14 |
| ProtoNet* [50] | Visformer-T | 62.48 ± 0.45 | 79.78 ± 0.30 |
| Meta-Baseline [7] | ResNet-12 | 63.17 ± 0.23 | 79.26 ± 0.17 |
| Meta-Baseline* [7] | Visformer-T | 62.59 ± 0.40 | 79.88 ± 0.35 |
| ours | Visformer-T | **76.71 ± 0.37** | **86.46 ± 0.27** |

the pre-training phase. Next, KCA is used in the meta-training phase, increasing the 1-shot and 5-shot accuracy by 3.19% and 3.3% This is because KCA can select class-specific features, progressively filtering out irrelevant features in a coarse-fine manner in three cascaded stages. Then, PCP is employed in the meta-tuning phase to learn pyramid prompts via deep interactions between modalities, which allows the ViT to dynamically modulate visual features based on semantic information, improving adaptability to spatial variations via the pyramid structure. Therefore, it is observed there are 8.24% and 2.72% improvements in the 1-shot and 5-shot tasks, respectively. Finally, when KCA and PCP together are incorporated into the baseline, a significant improvement of 11.14% and 4.84% can be observed, due to further capture of discriminative features guided by semantic information.

*4.4.2 **The effect of value $k$ in KCA**.* Table 4 shows the different values of $k$ in KCA. "$k = 1$" means that each query only selects one most relevant key to compute the attention matrix. smaller values of $k$ make it difficult to distill noisy tokens due to information interactions between irrelevant and relevant tokens. large values of $k$ make KCA tend to be a vanilla self-attention mechanism, both of which lead to performance degradation. For Visformer-T, $k = 10$

*4.4.3 **Backbone Architectures**.* Table 5 presents two baseline approaches based on Resnet-12 and corresponding reimplementations based on Visformer-T. Results are reported on miniImageNet. It is evident that directly replacing ResNet-12 with Visformer-T fails to improve results. Instead, our KTPP significantly surpasses baseline approaches with the identical Visformer Architecture due to select discriminative information and adaptively module visual features.

**Table 6: Comparison with state-of-the-art prompt and sparse attention methods on miniImageNet.**

|  | Method | 1-shot | 5-shot |
|---|---|---|---|
| Prompt | VPT [22] | 66.89 ± 0.35 | 83.42 ± 0.27 |
|  | SP [6] | 73.28 ± 0.40 | 85.23 ± 0.28 |
| Sparse attention | KVT [58] | 69.57 ± 0.44 | 84.78 ± 0.29 |
| ours | KTPP | **76.71 ± 0.37** | **86.46 ± 0.27** |

*4.4.4  Analysis of Prompt and Sparse Attention methods.* Table 6 shows the results of different Prompt and Sparse Attention methods. To fairly compare performance, we use the same ViT architectures in the FSL settings. We simply replace PCP with VPT [22] and SP [6] in the meta-tuning phase. However, VPT lacks semantic information by directly introducing some prompt parameters. Moreover, SP struggles to capture complex semantic relationships due to direct fusion. Instead, KTPP employs PCP to learn pyramid prompts, which allow the ViT to adaptively adjust visual features based on semantic information, making it robust to spatial variations via pyramid structure. Next, we directly replace KCA with KVT [58] in the whole training process. However, KVT has a poor generalization to new classes in sparse scenarios with only few labeled samples. Unlike KVT, we utilize semantic information to further extract discriminative visual features via cross-modal interaction in KCA. Meanwhile, the mean of all tokens as context prompts to enhance the global context information of each token. As a result, Our KTPP significantly outperforms the state-of-the-art prompt and sparse attention methods.

**Table 7: The effect of different classifiers on the FSL test.**

| Metrics for classifier | 1-shot | 5-shot |
|---|---|---|
| EU | 73.47 ± 0.35 | 83.53 ± 0.30 |
| CO | 76.33 ± 0.37 | 86.16 ± 0.30 |
| LR | **76.71 ± 0.37** | **86.46 ± 0.27** |

*4.4.5  Classifier Selection Strategy.* Table 7 shows three different classifiers and the results on miniImageNet are reported. Classification is achieved based on the distance between the support and query features in the feature space. Cosine distance to achieve nearest neighbor classification gets better results than Euclidean distance. Moreover, the Linear logistic regression classifier gets the best results due to constructing more support samples.

*4.4.6  Image Resolution.* Table 8 shows three common image resolutions in FSL. The ViT gets similar results at different resolutions, reflecting the adaptability of the proposed method to image resolution due to the full utilization of diverse spatial information.

*4.4.7  Visualization.* To qualitatively analyze the effectiveness of KTPP, we illustrate the t-SNE [54] results of baseline and KTPP. Fig. 5 shows the visualization of samples with novel classes on three benchmark datasets. In contrast to the spatially cluttered distribution of baseline, our KTPP allows the ViT to capture class-specific features so that intra-class images are closer together and inter-class images are further apart.

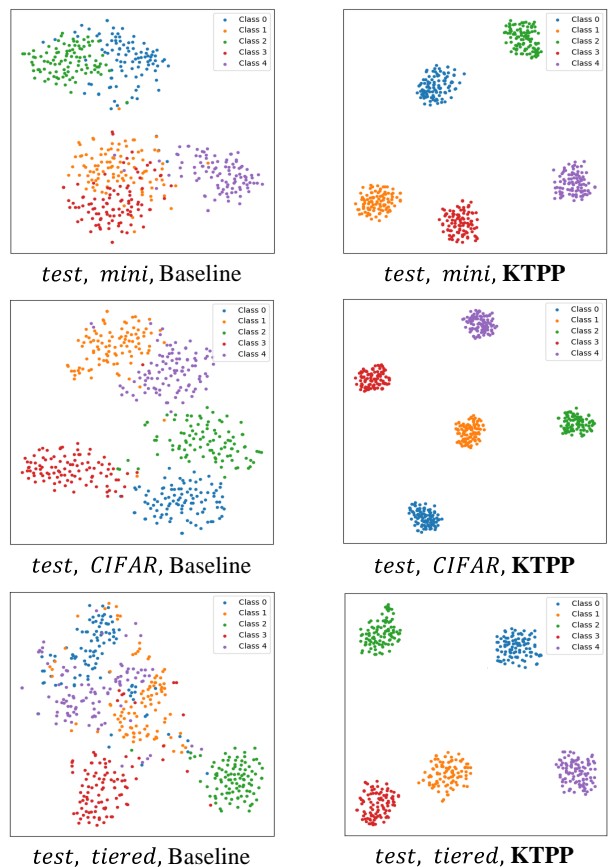

**Figure 5: T-SNE visualization results on novel classes from three benchmark datasets. ViT with KTPP performs better compared to ViT with baseline.**

**Table 8: The effect of different image resolutions.**

| Image resolution | 1-shot | 5-shot |
|---|---|---|
| 80 × 80 | 76.19 ± 0.40 | 86.02 ± 0.29 |
| 84 × 84 | 75.86 ± 0.40 | 85.86 ± 0.30 |
| 224 × 224 | **76.71 ± 0.37** | **86.46 ± 0.27** |

## 5  CONCLUSION

In this paper, we propose $k$-NN Transformer with Pyramid Prompts (KTPP) for Few-Shot Learning. KTPP mainly consists of $k$-NN Context Attention (KCA) and Pyramid Cross-modal Prompts (PCP). Specifically, KCA achieves progressively noisy filtration via a coarse-to-fine manner in the three cascaded stages and extracts discriminative features guided by semantic information. In PCP, pyramid prompts are generated to enhance visual features via deep cross-modal interactions between textual and multi-scale visual features. This enables the ViT to dynamically module the importance weights of visual features based on semantic information and make it robust to spatial variations via pyramid structure. Experimental results demonstrate the effectiveness of our KTPP model.

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
