# OpenReview forum: "KNN Transformer with Pyramid Prompts for Few-Shot Learning"
_acmmm.org/ACMMM/2024/Conference — MM2024 Poster_

### Official Review · Reviewer_p9wG · 2024-05-04

**Rating:** 5
**Confidence:** 3

**Summary:**

The paper proposes a KNN Transformer with pyramid prompts in few shot learning. In this approach, KNN is considered in attention mechanism, and pyramid is combined into prompt learning. Extensive experiments on four benchmark datasets demonstrate the effectiveness of proposed approach.

**Strengths:**

1. It is soundable for considering KNN into attention mechanism to filter out the background noise.
2. text feature is combined with different visual feature level by attention, it is a good idea, while the fusion stage should consider much more details.
3. The experiment is fine.

**Limitations:**

1. For example, as shown in fig. 1, the dog is the hot point in this figure. Since the KCA (combining KNN into attention) is done based on Vit, there is a question as follow.
How to really solve the problem of noise from the background? where  backgrounds in different images also have strong attention weights?
2. There is no cross domain dataset in experiment.
3. 'Figure 4: Illustration of cross-modal enhancement module in the PCA.' PCA is PCP?

**Suitability:**

3

---

### Official Review · Reviewer_UwsV · 2024-05-16

**Rating:** 5
**Confidence:** 4

**Summary:**

This paper proposes a k-NN Transformer with pyramid prompts, designed to select discriminative information through k-NN contextual attention and to adaptively modulate visual features using Pyramid Cross-modal Prompts (PCP). The cross-modal prompts are generated by CLIP's text encoder. The proposed method achieves promising results on several popular datasets.

**Strengths:**

1. This paper is well-written and easy to follow.
2. The proposed KCA attention mechanism, which filters tokens by calculating similarity, is novel. Additionally, the integration of CLIP's text encoder and the design of pyramid prompts assist in few-shot learning.
3. The experiments comprehensively demonstrate the effectiveness of the proposed method, showing significant improvements across various datasets.

**Limitations:**

1. The proposed method of designing prompts using CLIP's text encoder is complex. CLIP's text encoder already contains strong class priors, and directly adding a linear layer for training can also achieve good results, as demonstrated by SemFew[1].
2. Using auxiliary information to aid few-shot learning is only applicable to the ViT architecture and is not compatible with ResNet.
[1] Zhang, H., Xu, J., Jiang, S., & He, Z. Simple Semantic-Aided Few-Shot Learning. CVPR2024.

**Suitability:**

3

---

### Official Review · Reviewer_fKBP · 2024-05-25

**Rating:** 3
**Confidence:** 3

**Summary:**

This paper addresses the problem of few-shot learning. The authors first propose an improved attention calculation module, KCA, to enable the model to focus only on relevant tokens. Additionally, they introduce language knowledge into the model framework through the CLIP model and design corresponding knowledge fusion modules (PcP and Fusion).  On multiple benchmarks for this task, the proposed method achieves state-of-the-art performance.

**Strengths:**

1.  The motivation is reasonable, and the specific methods closely align with the motivation.
2.  The writing and illustrations are relatively clear and easy to understand.
3. The proposed modules are complex and detailed, reflecting the authors' deep thinking.

**Limitations:**

1. The KCA module, which constructs the attention map based on the top k keys, seems to have already been proposed [2]. Therefore, regarding the improvement of attention, this might not provide much additional insight. Additionally, in the ViT or transformer community, there are already many improved self-attention methods that use more sparse attention to enhance model performance or accelerate model convergence, such as Deformable Attention in Deformable DETR. It seems that the authors did not discuss or compare these methods in the paper.
2. The main experiments lack sufficient comparisons. Some recent methods that use language knowledge or the CLIP module have not been compared, such as [1].
3.  The ablation study of the PCP module and its related context is not thorough enough. In the paper, the authors conducted an ablation study on the entire PCP module, which seems to be too coarse-grained. First, the Fusion module behind stage3 and the PcP module takes 𝑍^{S3}{v}, 𝑍^{fp}{t}, 𝑍^{cp}{t} as inputs. Does it make sense to use these inputs and combine them? For example, what would happen to the performance if 𝑍^{cp}{t} is not concatenated to the tokens? Second,  PCP calculates prompts 𝑍^{ppi}{t} for each stage module. Does the prompts for each stage help the subsequent model process? I think the authors can try ablation studies on prompts 𝑍^{ppi}{t} outputted by different stages.
4. Some expressions in the paper are not very clear. For example, how is 𝑍^{cp}{t} obtained from the text encoder of CLIP? How is L{t} obtained?
5. Some typo. For example, in Eqn 6, A_{ij} should be A; in line 541, q_{i} should be a support image.

While this paper still has these limitations, I believe the authors have put in a lot of effort to complete this work. If the authors can supplement the experiments regarding the second and third point, I would consider giving it a higher score.

[1] Self-Prompt Mechanism for Few-Shot Image Recognition
[2] Mixed Autoencoder for Self-supervised Visual Representation Learning

**Suitability:**

2

---

### Official Review · Reviewer_m11x · 2024-05-27

**Rating:** 5
**Confidence:** 3

**Summary:**

For Few-Shot Learning, this paper proposes a 𝑘-NN Transformer with Pyramid Prompts (KTPP), which involves two novel modules named 𝑘-NN Context Attention (KCA) and Pyramid Cross-modal Prompts (PCP). Specifically, KCA achieves progressively noisy filtration via a coarseto-fine manner in the three cascaded stages and extracts discriminative features guided by semantic information. In PCP, pyramid prompts are generated to enhance visual features via deep crossmodal interactions between textual and multi-scale visual features. Experimental results show that the KTPP model is effective.

**Strengths:**

1. Paper is easy to follow, and motivation is clear.
2. The proposed 𝑘-NN Context Attention (KCA) and Pyramid Cross-modal Prompts (PCP) are interesting, and are effective for few-shot learning according to the experimental results.
3. The experimental results are good, the proposed KTPP model achieves new state-of-the-art performance.

**Limitations:**

1. In related work section, please add more discussion about the relevant works of CLIP for few-shot learning.
2. The Fusion module in Figure 2, fuses three features including the support features $𝑍_𝑣^𝑆$, text-based class-aware prompts $𝑍_𝑡^{𝑐𝑝}$, and pyramid prompts $𝑍_𝑡^{𝑝𝑝}$. Please provide more detail ablation study to explore the effectiveness of these three features. For example, $𝑍_𝑣^𝑆$, $𝑍_𝑣^𝑆$ + $𝑍_𝑡^{𝑐𝑝}$, $𝑍_𝑣^𝑆$ + $𝑍_𝑡^{𝑝𝑝}$.

**Suitability:**

2

---

### Meta-Review · Area_Chair_hQmp · 2024-07-02

**Recommendation:** Accept (Poster)
**Confidence:** 4

**Metareview:**

The KNN Transformer idea was deemed as quite interesting and new. The results are strong.
The author rebuttal was effective, which served to convince one reviewer to improve his/her rating (and that reviewer's initial comments were quite positive too, despite the initial rating). Overall, it should be a good paper to include for the conference.